# DimOL: Dimensional Awareness as A New 'Dimension' in Operator Learning

## Abstract

In the realm of computational physics, an enduring topic is the numerical solutions to partial differential equations (PDEs). Recently, the attention of researchers has shifted towards Neural Operator methods, renowned for their capability to approximate "operators" — mappings from functions to functions. Despite the universal approximation theorem within neural operators, ensuring error bounds often requires employing numerous Fourier layers. However, what about lightweight models? In response to this question, we introduce DimOL (Dimension-aware Operator Learning), drawing insights from dimensional analysis. To implement DimOL, we propose the ProdLayer, which can be seamlessly integrated into FNO-based and Transformer-based PDE solvers, enhancing their ability to handle sum-of-products structures inherent in many physical systems. Empirically, DimOL models achieve up to $48\%$ performance gain within the PDE datasets. Furthermore, by analyzing Fourier components' weights, we can symbolically discern the physical significance of each term. This sheds light on the opaque nature of neural networks, unveiling underlying physical principles.

## 1 Introduction

The Neural Operator methods have come to vision within the field of AI for science (AI4Sci), for their excellent property of having the ability to approximate any operators (*i.e.* mapping from function space to another function space), while traditional neural networks would fail.

However, are these models "scientific" enough for AI4Sci? By expanding the TorusLi dataset to TorusVisForce dataset, the F-FNO paper (Tran et al., 2023) has raised a crucial question on how the Neural Operator methods should perform when we hope the model could handle multiple background situations (such as different viscosities in fluid dynamic problems), which would require a deeper comprehension of physics. However, the F-FNO paper still did not discuss this issue thoroughly: It lacks a comparative analysis of F-FNO model performance against other models, and notably, F-FNO itself does not incorporate dimension awareness.

This research would reveal one simple but profound fact: *Models equipped with dimensional awareness tend to perform universally better.* Accordingly, we propose Dimension-Aware Operator Learning (DimOL) and see how to improve models dealing with physical quantities, including FNO-based methods such as FNO (Li et al., 2021), F-FNO (Tran et al., 2023), T-FNO (Tran et al., 2023), and non-FNO methods such as the Transformer-based LSM (Wu et al., 2023). Our research indicates that dimension-aware models demonstrate improved performance as the variety of physical quantities provided to the model increases. Specifically, within the TorusVisForce dataset proposed by F-FNO (Tran et al., 2023) (which involves three physical quantities), dimension-aware models can reduce the prediction error by up to $48\%$ compared to the backbone models across both full-scale dataset and few-shot tasks.

Building upon the widely recognized physical intuition that quantities with different units (or dimensions) cannot be directly added but can be sensibly multiplied, all of the improvements above are achieved by encoding product terms between channels using a novel ProdLayer — a simple plugin-and-use module designed in this work. Furthermore, we conduct comparisons between the ProdLayer and alternative approaches such as quadratic layers, which theoretically could serve similar roles. However, our findings demonstrate that the ProdLayer outperforms these alternatives.

In this work, we also demonstrate that dimensional awareness is not merely about randomly adding some product term encoding to the base model. Empirically, most adaptations are effective only when we consider the supposed dimensions of the latent vectors. For example, in FNO-based models, replacing the linear transform on the Fourier domain with a ProdLayer does not yield improvement. This approach introduces a global convolution on the variable field ($\mathcal{F}^{-1}(\hat{f}\hat{g}) = f * g$), which lacks physical significance in this context. Therefore, to adapt a model to incorporate DimOL, one must carefully consider the physical meaning of the model's latent space and proceed accordingly.

We evaluate DimOL on classic physics simulation datasets, including regular mesh scenarios (Burgers, Darcy Flow, and Navier-Stokes) and irregular mesh settings (Inductor2D). We showcase the effectiveness of DimOL compared to existing FNO-based and Transformer-based models.

Notably, quantifying and interpreting the profits of considering dimensional awareness as a property of a model remains an open problem. One approach is to apply the model to simpler, more comprehensible datasets, such as the evolution operator for the Burgers equation in Section 4.7. This allows for identifying which dimensions are associated with the ProdLayer by analyzing the Fourier weights. However, for more complex datasets, the relationships among the Fourier weights may not have clear symbolic interpretations, making it difficult to analyze their internal connections. Another way to investigate the benefits of introducing dimensional awareness involves assessing the model's accuracy on similarly transformed datasets, as discussed in Section 4.6. It is more appropriate to view dimensional awareness as a novel and effective design principle for neural operators, rather than as a definitive scientific claim about a model's capacity to perceive dimensions.

Another advantage of integrating dimensional awareness with operator learning—consistent with our scientific common sense—is its potential to reverse-engineer the meanings of latent variables and discover their symbolically meaningful relationships. This ability is effectively demonstrated through the experiments presented in this paper.

## 2 RELATED WORK

### 2.1 NUMERICAL PDE SOLVERS

Traditional methods (Grossmann, 2007; Ŝolín, 2005; Ciarlet, 2002; Courant et al., 1967; Cooley et al., 1969; Gottlieb & Orszag, 1977; Fornberg, 1998; Kopriva, 2009) for addressing partial differential equation (PDE) systems encompass techniques like finite element methods (FEMs), finite difference methods (FDMs), finite volume methods (FVMs), and pseudo-spectral methods, including Crank-Nicholson and Carpenter-Kennedy. These approaches typically involve discretizing the spatial domain, where achieving higher precision requires finer discretizations, leading to greater computational demands. To manage these costs, historically, specific PDEs have been addressed using streamlined models such as Reynolds averaged Navier-Stokes and large eddy simulations. In recent developments, machine learning has emerged as a viable option for expediting simulations.

### 2.2 MACHINE LEARNING METHODS FOR PDES

**Equation-constrained methods.** These methods usually follow the PINN (Raissi et al., 2019; Lu et al., 2021b; Karniadakis et al., 2021) paradigm, where they move all the terms of constraining equations to the left side, square left-hand-side up, and take it as the objective function (Yu et al., 2018; Wang et al., 2022; 2021). However, This method does not generalize well on different boundary conditions (BC) and initial conditions (IC). Another fatal drawback is that you can only apply this method with the knowledge of the dynamic system, including the dynamic equations, BC, and IC, but one major effect of machine learning is to learn from real-world data, which is not feasible for equation-constrained methods. Thus, this paper will focus on Neural Operator methods and other state-of-the-art Transformer-based methods instead.

**Neural operators.** Operator Learning has recently become a heated topic within AI for Science. There are mainly two branches of Operator Learning methods: One is based on DeepONet (Lu et al., 2021a), and the other is based on Fourier Neural Operator (FNO) (Li et al., 2021), both of which have been mathematically proven to have the capability to approximate any operators. DeepONet constructs a branch network and a trunk network derived from the universal approximation theorem (Chen & Chen, 1995) to process the input functions and query points, while FNO utilizes discrete Fourier transforms to approximate continuous Fourier transform, which can represent kernel

integral operators. FNO can also gain a remarkable trade-off between cost and accuracy in practice due to its quasi-linear time complexity. So in this paper, we would mainly discuss FNO-based methods (Tran et al., 2023; Kossaifi et al., 2023; Xiong et al., 2022; Wen et al., 2022; Ashiqur Rahman et al., 2022) for Neural Operators. Other typical FNO-based methods include: Factorized Fourier Neural Operator (F-FNO) (Tran et al., 2023) and Tensorized Fourier Neural Operator (T-FNO) (Kossaifi et al., 2023). F-FNO is based on doing 1D Fourier transforms along axes respectively to reduce the number of parameters, while T-FNO focuses on using tensor factorizations (such as Tucker factorization). However, these works are not scalable to handle problems with multiple types of input functions. Another line of work (Cao, 2021) proposes to use the attention mechanism (Vaswani et al., 2017) for learning operators.

**Transformer-based methods.** Transformers (Katharopoulos et al., 2020; Schlag et al., 2021; Xiong et al., 2020; 2021; Shen et al., 2021; Nguyen & Salazar, 2019) have been long proven to be an effective module in many machine learning tasks. Recent research has shown that this is also true for physics-simulation tasks. Latent Spectral Method (LSM) (Wu et al., 2023) uses sine functions to approximate any function guaranteed by the theorem of Convergence of Trigonometric Approximation (Dyachenko, 1995), and utilizes an U-Net structure (Ronneberger et al., 2015) to handle multi-scale problems. General Neural Operator Transformer (GNOT) (Hao et al., 2023), on the other hand, designed a novel heterogeneous normalized attention layer to handle the grids and input functions flexibly. Both of them have remarkable performances, but we are still not sure whether they can approximate any operators like DeepONet and FNO do (instead of merely approximating functions), so we put them in the category of Transformer-based methods.

## 3 METHODOLOGY

In this section, we present the technical details of a simple yet effective approach named dimension-aware operator learning (DimOL), which can be seamlessly integrated into a wide range of neural operator-based PDE solvers. In Section 3.1, we initially introduce the general intuition behind DimOL. In Section 3.2, we propose a novel and lightweight neural network module named ProdLayer, which is motivated by dimensional awareness. In Section 3.3 and Section 3.4, we provide two specific implementations of using ProdLayer to replace the original neural network blocks in existing models, showing that DimOL is a universal technique applicable to various neural operator learning methods, adding negligible computational overhead.

### 3.1 INTUITION OF DIMENSIONAL AWARENESS

Before introducing a specific model, it is important to understand what dimensional awareness is and why we believe it is crucial in solving partial differential equations (PDEs) in physical systems.

Dimensional awareness is a methodology that involves recognizing and incorporating the intrinsic dimensions and units of physical quantities into the design of AI models. This approach ensures that the model respects the physical laws governing the system, leading to more accurate and reliable predictions. Consequently, we believe that dimensional awareness would be a new dimension that we must consider when designing a new model for physical simulation.

In the context of solving PDEs, dimensional awareness allows the model to explicitly encode relationships and interactions between different physical quantities. For instance, in many physical systems, variables are often related through products and ratios that have specific dimensional properties. Traditional AI models may overlook these crucial relationships, leading to suboptimal results. In contrast, by encoding these product terms into the neural network, the model can better capture the underlying physics of the system, resulting in improved performance and generalization.

Below, we will use an example to briefly demonstrate why simply encoding product terms would improve model performance. Let's consider a classic prediction task in physics: predicting the future state of a physical field given its current state. The goal is to input a field of a physical quantity at a given moment and predict the same field after a certain period. This system is governed by a dynamic equation, often expressed as:

$$\frac{d}{dt}\mathbf{u}(\mathbf{x}, t) = \mathcal{G}\mathbf{u}(\mathbf{x}, t), \tag{1}$$

where $\mathcal{G}$ is the time-evolution operator. In real-life situations, $\mathcal{G}$ frequently appears as a summation of products, with each product term involving physical quantities of different dimensions. Despite the

differing dimensions of the quantities involved, the resulting product terms share the same dimension as $\frac{d}{dt}\mathbf{u}(\mathbf{x}, t)$. For instance, consider the Navier-Stokes equations (Temam, 2001), which describe the motion of fluid substances. The governing equation for the Navier-Stokes problem is:

$$\rho\frac{d}{dt}\boldsymbol{v} = -\rho\boldsymbol{v} \cdot \nabla\boldsymbol{v} - \nabla p + \mu\nabla^2\boldsymbol{v} + \rho\boldsymbol{f}, \tag{2}$$

where $\rho$ is the fluid density, $\boldsymbol{v}$ is the velocity field, $p$ is the pressure field, $\mu$ is the dynamic viscosity, and $\boldsymbol{f}$ is the external force field. In this equation, the left-hand side $\rho\frac{d}{dt}\boldsymbol{v}$ represents the rate of change of momentum per unit volume. On the right-hand side, we see a summation of terms involving products of physical quantities, where $\rho\boldsymbol{v} \cdot \nabla\boldsymbol{v}$ is the advection term, representing the transport of momentum, $\nabla p$ is the pressure gradient term, $\mu\nabla^2\boldsymbol{v}$ is the diffusion term, and $\rho\boldsymbol{f}$ is the external force term. Each term on the right-hand side involves different combinations of physical quantities, yet all terms conform to the dimension of force per unit volume, matching the left-hand side.

Classic high-accuracy numerical methods, such as implicit Runge-Kutta methods, are often used to approximate time evolution in physical systems. These implicit methods involve numerous iterative steps to minimize the error within an acceptable tolerance, making them much slower compared to learning-based methods. On the other hand, explicit numerical methods, like the Forward Euler and explicit Runge-Kutta methods, are much faster. These methods involve straightforward calculations of the time-evolution operator $\mathcal{G}$ for a finite number of times. Although explicit methods are computationally efficient, they can be less accurate and stable compared to implicit methods, especially for stiff equations or long-term integration. With our basic intuition, we would expect our neural operator learning method to at least have the capability to simulate explicit numerical methods effectively. Fortunately, the neural networks could do better than that since the training dataset is generated with implicit methods and even real-world data. As a result, while the models might simulate explicit methods in computational efficiency, they can also achieve the high accuracy and robustness typically associated with implicit methods.

Now, we want to go one step further. If the objective of the learning-based models is to perform well across a wide range of input data distributions, the most straightforward approach is to enable the model to directly regress polynomial-structured terms. If the model has done it right, it should be robust to variations in input scale and distribution. Our research demonstrates that dimension-aware models significantly outperform traditional models under such conditions. These models are inherently better at maintaining accuracy even when inputs are scaled by large constants.

Another intuition comes from the Separation of Variables Method, a powerful technique commonly used in PDE analysis. Let's consider the 1D heat equation as an example:

$$\frac{\partial u}{\partial t} - \alpha\frac{\partial^2 u}{\partial x^2} = 0. \tag{3}$$

When using the Separation of Variables method, the solution can be expressed as:

$$u(x, t) = \sum_{n=1}^{\infty} D_n X(x)T(t) = \sum_{n=1}^{\infty} D_n \sin\frac{n\pi x}{L} \exp\left(-\frac{n^2\pi^2\alpha t}{L^2}\right). \tag{4}$$

This solution exactly takes the form of a sum-of-products, where $X(x)$ and $T(t)$ are the spatial and temporal components, respectively, and $D_n$ are the coefficients. This form is highly amenable to learning-based approaches, as it suggests that the model can effectively learn to regress these coefficients and basis functions.

### 3.2 PRODLAYER: INTEGRATING DIMENSION-AWARE PRODUCT TERMS

To encode product terms in a neural network, the most straightforward approach is to directly multiply two terms. This leads us to the design of the ProdLayer, which can be integrated into the model to handle such operations efficiently. Let's define the input vector $\mathbf{x}$ as follows:

$$\mathbf{x} = \mathbf{x}_a \oplus \mathbf{x}_b \oplus \mathbf{x}_2, \tag{5}$$

where $\oplus$ denotes feature concatenation, $\mathbf{x}_a$ and $\mathbf{x}_b$ are sub-vectors with the same dimensionality, *i.e.*, $\dim(\mathbf{x}_a) = \dim(\mathbf{x}_b) = p$, and $p$ denotes the number of product terms.

The ProdLayer operates as follows:

$$\sigma_{\text{prod}}(\mathbf{x}) = W\left[(\mathbf{x}_a \otimes \mathbf{x}_b) \oplus \mathbf{x}_2\right], \tag{6}$$

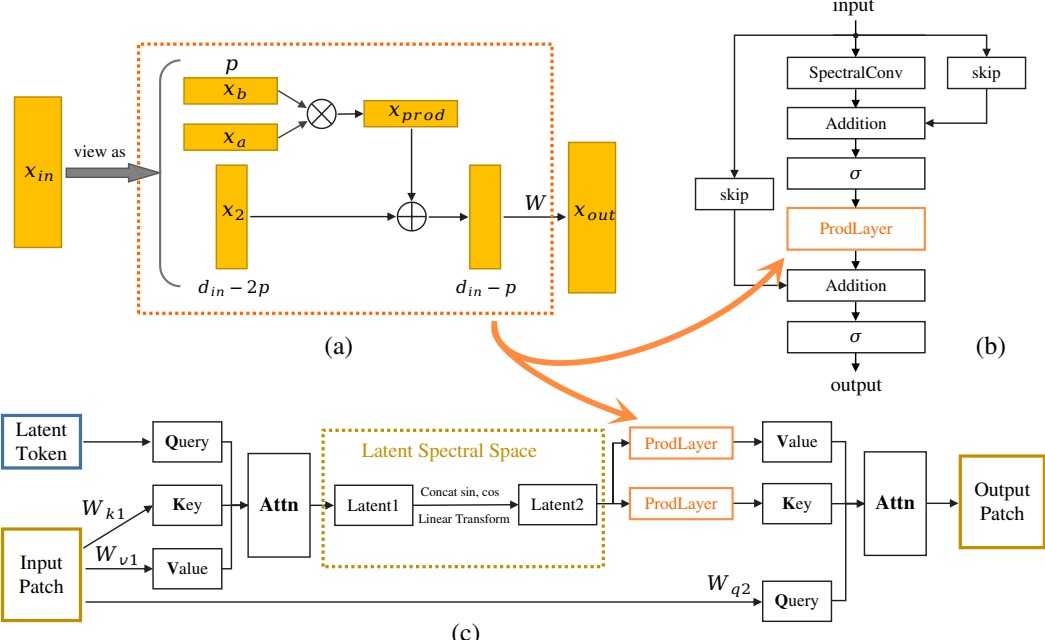

Figure 1: An illustration of our approach: **(a)** Structure of the ProdLayer. This layer directly encodes product terms by element-wise multiplication of $\mathbf{x}_a$ and $\mathbf{x}_b$, followed by concatenation with $\mathbf{x}_2$ and a linear transformation. **(b&c)** We can seamlessly integrate the ProdLayer in existing FNO-based models (as a substitution of the original MLPs) and Transformer-based models like LSM (as additional modules with negligible extra parameters).

where $\otimes$ denotes the element-wise product and $W$ represents a learnable weight matrix. This formulation allows the layer to directly incorporate the product terms of $\mathbf{x}_a$ and $\mathbf{x}_b$, combined with additional input features $\mathbf{x}_2$. The structure of the ProdLayer is shown in Figure 1(a).

Another simple way to introduce product terms is by using the identity $ab = \frac{1}{4}[(a+b)^2 - (a-b)^2]$, which expresses the product as the difference of two square terms. However, within the TorusLi experiment, as shown in Table 3, this indirect method of encoding product terms is demonstrated to be less effective than the ProdLayer.

The mechanism of the ProdLayer allows the neural network to capture and encode the product terms directly, leveraging the physical intuition behind polynomial structures and product terms found in many dynamic systems governed by PDEs. By incorporating the ProdLayer, the model can effectively learn and represent these relationships, improving its performance on tasks involving a wide range of input data distributions.

## 3.3 FNO-BASED MODELS WITH PRODLAYER

For FNO-based models, introducing product-term encoding via the ProdLayer enables the model to approximate explicit numerical methods more accurately. This is particularly beneficial when the evolution operator takes the form of sum-of-products, a common scenario in complex dynamic systems. As illustrated in Figure 1(b), we choose to replace the MLP layers of the original FNO with the ProdLayer. This modification allows the model to directly encode product terms, enhancing its ability to capture the intrinsic relationships between physical quantities.

In our implementation, we adopt the backbone of the T-FNO. This choice offers the flexibility to switch between using tensor factorization and the original FNO, providing a versatile framework for experimentation. When tensor factorization is not used, the model functions as a standard FNO. We demonstrate the effectiveness of our approach by presenting uniformly enhanced performance with the ProdLayer. By analyzing the Fourier weights, we provide insights into how the network conceptualizes the physical quantities involved. This analysis is particularly enlightening in the context of a toy example related to the Burgers equation, discussed in Section 4.7.

Table 1: Results on the Burgers dataset, where *chan. mix* denotes the use of a channel mixing layer.

| Model | # Parameters (Million) | MSE ($\times 10^{-3}$) | Gain |
|---|---|---|---|
| FNO w/o *chan. mix* (Li et al., 2021) | 0.098 | 0.646 | - |
| FNO w/ *chan. mix* | 0.106 | 1.045 | -61.8% |
| FNO + ProdLayer | 0.105 | 0.593 | 8.2% |
| T-FNO w/o *chan. mix* (Li et al., 2021) | 0.052 | 2.225 | - |
| T-FNO w/ *chan. mix* | 0.056 | 2.342 | -5,3% |
| T-FNO + ProdLayer | 0.056 | 1.147 | 48.4% |

### 3.4 TRANSFORMER-BASED MODELS WITH PRODLAYER

The Latent Spectral Method (LSM) is a cutting-edge Transformer-based model designed for solving PDEs. Its basic innovation is the incorporation of Sine functions into the network, allowing it to approximate any (periodic) function within a specific spectral range. For non-periodic components, LSM utilizes convolution-based neural blocks. The overall structure of LSM is akin to a U-Net, enabling it to learn interactions between different patches of the domain at varying scales.

Inspired by the separation of variables method, we hypothesize that the projection in the latent spectral space can be enhanced if terms can multiply Sine or Cosine functions directly, rather than merely adding them as biases. This modification aligns with the core principle of DimOL, which aims to improve model performance by encoding product terms. To implement this idea, we replace the linear transformation in the decoder of LSM with a ProdLayer, as shown in Figure1(c). This enhanced architecture retains the original strengths of LSM while integrating the ProdLayer to facilitate improved spectral projections, allowing the model to handle product terms more effectively. Given this intuition, a larger number of product terms for each ProdLayer is beneficial and can enhance the model's capacity to capture complex interactions.

## 4 EXPERIMENTS

### 4.1 EXPERIMENTAL SETUPS

**Datasets.** We apply DimOL to multiple base models and evaluate their performance in a variety of settings, including both regular mesh settings (Burgers, Darcy Flow, Navier-Stokes) and irregular mesh settings (Inductor2D). Notably, DimOL does not apply to geometry-only datasets. In these datasets, the inputs consist solely of coordinates, which lack the physical dimensions that DimOL leverages. Thus, there is no dimensional information for DimOL to exploit. However, this does not imply that DimOL would fail on irregular meshes. The GNOT paper introduced the Inductor2D dataset, which includes physical quantities as part of the input. This allows us to test the model's generalization ability to real-world scenarios where background physical conditions could change.

**Compared models.** For the Burgers, Darcy Flow, and Navier-Stokes datasets, we test DimOL on several neural operator models, including FNO, F-FNO, T-FNO, and LSM. These models were chosen due to their effectiveness in handling PDEs across different settings. As for Inductor2D, we adapt the GNOT model with DimOL. The models we use are 4-layer models.

**Hyperparameter.** The following experiments use a product number of $p = 2$ in each ProdLayer (see Figure 1). While this empirical value may not be the optimal solution for every specific dataset, it consistently leads to improvements in our experiments. Other options for the $p$ value also yield performance improvements and are discussed in the ablation study in Section 4.8. Specifically, setting $p = 0$ would reduce the channel-mixing layer of the model to an ordinary MLP. For FNO and LSM models, we adopt the original depth settings of 4 layers.

### 4.2 BURGERS

We adopt the Burgers dataset of the FNO paper. The 1D Burgers' equation is given by $\partial_t u(x,t) + u(x,t)\partial_x(u(x,t)) = \nu \partial_{xx} u(x,t), \ x \in (0,1). t \in (0,1]$. Since this is a 1D problem, more complex methods like F-FNO, T-FNO, and LSM are not applicable, so we compare FNO with its dimension-aware counterpart, FNO + ProdLayer. As shown in Table 1, we evaluate the 4-layer T-FNO models

Table 2: Model performance on the Darcy Flow dataset. Notably, the use of DimOL provides a consistent performance improvement to all compared models.

| Model | # Parameters (Million) | MSE ($\times 10^{-3}$) | Gain |
|---|---|---|---|
| FNO (Li et al., 2021) | 1.658 | 5.889 | - |
| FNO + ProdLayer | 1.663 | 5.726 | 2.8% |
| T-FNO (Kossaifi et al., 2023) | 0.684 | 7.500 | - |
| T-FNO + ProdLayer | 0.688 | 6.919 | 7.8% |
| LSM (Wu et al., 2023) | 4.813 | 2.892 | - |
| LSM + ProdLayer | 4.812 | 2.713 | 6.2% |

Table 3: Model performance on the TorusLi dataset.

| Base model | DimOL | MSE | Gain |
|---|---|---|---|
| FNO (Li et al., 2021) | - | 0.1496 | - |
| FNO | QuadLayer | 0.1431 | 4.3% |
| FNO | ProdLayer | 0.1394 | 6.8% |
| T-FNO (Kossaifi et al., 2023) | - | 0.1448 | - |
| T-FNO | QuadLayer | 0.1430 | 1.2% |
| T-FNO | ProdLayer | 0.1404 | 2.3% |
| F-FNO (Tran et al., 2023) | - | 0.1651 | - |
| F-FNO | ProdLayer | 0.1542 | 6.6% |

on the Burgers dataset and achieve an impressive $48.4\%$ drop in MSE on the test set. We also compare the model with the version with channel-mixing of MLP, which would overfit and have lower performance. In contrast, the models with ProdLayers as channel-mixing are less likely to overfit, due to its intrinsic dimensional awareness.

## 4.3 DARCY FLOW

This dataset is also sourced from FNO. The physical process of Darcy Flow can be described by the equation $-\nabla \cdot (a(x)\nabla u(x)) = f(x), \ x \in (0,1)^2$, where the input is $a(x)$ and the output is $u(x)$, which satisfies the steady-state equation. Its important to note that this setting focuses on finding a steady state under given conditions, differing from dynamic prediction tasks. As shown in Table 2, the DimOL-adapted models consistently outperform the original versions.

## 4.4 NAVIER-STOKES

The TorusLi dataset is introduced by the original FNO paper. The governing equation is as follows: $\frac{\partial \omega}{\partial t} + \mathbf{u} \cdot \nabla \omega = \nu \nabla^2 \omega + f, \ \nabla \cdot \mathbf{u} = 0, \ \nabla \times \mathbf{u} = w\mathbf{e_z}$. The input is the initial vorticity $\omega_0(x,y) = \omega(x,y,t=0)$, while the external force term is fixed within the dataset. A main drawback of the TorusLi dataset: is lacking the diversity of other variables. To resolve this, F-FNO (Tran et al., 2023) introduces the TorusVisForce dataset. The core idea is to ensure that the model is general enough to handle varying environmental conditions, such as different external force terms and viscosity coefficients. To achieve this, these terms are set as random during dataset generation and are used as inputs for the model. Additionally, FNO conducts TorusLi experiments using a sequence of previous fluid fields (*e.g.*, 10 steps). F-FNO suggests that this task should adhere to the first-order Markov property, meaning that one time step is sufficient. We adopt this setting, which may explain why LSM does not outperform FNO methods here, contrary to the findings reported in the LSM paper.

**Results on TorusLi.** As shown in Table 3, DimOL models consistently outperform the base models on the TorusLi dataset. Additionally, we test the product term encoding with quadratic layers (*i.e.*, ProdLayer). However, as demonstrated in the tests on FNO and T-FNO, the quadratic layer does not surpass the performance of the more straightforward ProdLayer.

**Results on TorusVisForce.** Due to the size of the TorusVisForce dataset, we select T-FNO for our case study, given its strong performance on TorusLi. In the TorusVisForce dataset, the involvement

Table 4: Model performance on the TorusVisForce dataset with different prediction horizons.

| Model | $T = 4$ | | $T = 10$ | |
| | MSE ($\times 10^{-2}$) | Gain | MSE ($\times 10^{-2}$) | Gain |
| --- | --- | --- | --- | --- |
| T-FNO (Kossaifi et al., 2023) | 1.090 | - | 1.717 | - |
| T-FNO + ProdLayer | 0.966 | 11.5% | 1.622 | 5.5% |

Table 5: Prediction errors on the Inductor2D dataset with irregular meshes.

| Model | MSE-$A_z$ ($\times 10^{-2}$) | MSE-$B_x$ | MSE-$B_y$ | Gain |
| --- | --- | --- | --- | --- |
| GNOT (Hao et al., 2023) | 0.8736 | 0.2579 | 0.3873 | - |
| GNOT + ProdLayer | 0.8869 | 0.2578 | 0.3847 | 4.8% |

of more physical quantities in the input allows the DimOL models to achieve significant improvements. We conduct two tasks: in the first, we set the prediction horizon to $T = 4$ time steps, while in the second, we have $T = 10$. Obviously, as the prediction horizon extends, the task becomes more challenging. Table 4 presents the prediction errors. We can observe that our DimOL method significantly improves the performance of the base models in both tasks.

## 4.5 INDUCTOR2D

The datasets mentioned above primarily involve regular meshes. What about irregular meshes? One might suggest checking out Geo-FNO's Airfoil, Pipe, and Elasticity datasets. However, does dimensional awareness truly apply here? These datasets are mesh-only and lack any physical quantities as inputs! Moreover, the datasets are pretty weird by making all the meshes outside the boundaries also a part of the input. They're merely redundant "auxiliary lines" that are used to generate the dataset, and no matter how they twist, the boundary would remain the same. This can confuse the model, leading to inconsistent outputs. Therefore, we adopt the Inductor2D dataset introduced by GNOT.

In our implementation, the adaptation with GNOT is similar to how we adapted LSM, specifically by replacing the decoder's linear projections with ProdLayers. We do not perform any hyperparameter search. The model follows the default hyperparameters of GNOT: the hidden dimension is $64$, and the number of attention heads is set to $8$. As illustrated in Table 5, we achieve improved results by simply applying DimOL with the number of products set to $p = 1$.

## 4.6 FEW-SHOT LEARNING STUDIES ON OPERATOR PROPERTIES

We aim to develop a model that approximates an operator while preserving key properties of the original operator. If successful, the model should generalize well in few-shot learning scenarios. To evaluate this, we design three experimental settings:

- *Datasets*: We construct two subsets of the TorusVisForce dataset, named TorusVisForce-*TopMu* and -*LowMu*. These datasets contain 200 and 100 data sequences respectively with the highest and lowest viscosity values ranging in $[8.2 \times 10^{-5}, 1 \times 10^{-4}]$ and $[1 \times 10^{-5}, 1.8 \times 10^{-5}]$.

- *Few-shot training*: We use 100 random samples from TorusVisForce-*TopMu* as the training set.

- *In-set testing*: We first perform in-distribution testing on the other partition of 100 unseen training samples from the TorusVisForce-*TopMu* subset.

- *Partial-OOD testing*: We generate a partially OOD test set by scaling the training data using a coefficient $k$. Here, $k$ is an integer applied to the physical quantities $\omega$, $f$, and $\nu$.

- *OOD testing*: We finally use TorusVisForce-*LowMu* as the test set to evaluate the model's generalization ability to OOD data.

For *Partial-OOD* testing, we here leverage a crucial principle from dimensional analysis: the invariance of similar transformations. This principle motivates us to create *partially out-of-distribution* test data using simple data transformations to assess operator learning. The transformed data, while indeed outside the original data distribution, effectively maintains similar dynamics to the training data. We use the TorusVisForce dataset as a case study. Consider data transformations:

$$\omega_0 \to k\omega_0, \quad t \to t/k, \quad \nu \to k\nu, \quad f \to k^2 f. \tag{7}$$

Table 6: Prediction errors ($\times 10^{-2}$) on TorusVisForce ($T = 4$) in few-shot learning and OOD setups.

| Base | Method | MSE (In-Set) | MSE ($k = 4$) | MSE ($k = 16$ ) | MSE (OOD) |
|------|--------|:---:|:---:|:---:|:---:|
| T-FNO | w/o *chan. mix* | 2.354 | 1.158 | 1.246 | 8.975 |
| T-FNO | $p = 0$ | 1.625 | 1.005 | 1.062 | 8.000 |
| T-FNO | $p = 2$ | 1.456 | 0.911 | 0.984 | 7.828 |
| | Gain | 10.4% | 9.4% | 7.3% | 2.2% |
| LSM | $p = 0$ | 8.304 | 1.427 | 1.500 | 16.390 |
| LSM | $p = 2$ | 8.031 | 1.262 | 1.334 | 16.040 |
| | Gain | 3.3% | 11.6% | 11.1% | 2.1% |

If we scale the inputs, the output should adhere to the following relationship:

$$\omega_{\text{new}}(x, T_{\text{new}} = T/k) = k\omega(x, T), \tag{8}$$

which is validated using a traditional numerical solver, although a slight numerical bias is observed. Therefore, in the *Partial-OOD* experiments, although DimOL cannot access the analytic expression for the actual operator $G^{\dagger} : [H_{per}^r((0,1)^2), L_{per}^2((0,1)^2; \mathbb{R}), \mathbb{R}_+] \to H_{per}^r((0,1)^2$ in TorusVisForce, it will seek to uncover the key properties from the transformed data.

Table 6 reveals the impressive few-shot learning capability of DimOL models. More significantly, it highlights the acquisition of a similar-transformation invariance property, which scientists may anticipate from the ground-truth operator. This property is particularly beneficial when combining machine learning with traditional dimensional analysis.

### 4.7 THE POTENTIAL OF FINDING NEW PHYSICAL TERMS WITH DIMOL

Symbolic regression has long been a prominent topic in AI for Science, yet combining symbolic methods with neural networks remains a challenge. Symbolically regressing a specific function is already complex, let alone regressing operators. However, by incorporating dimensional awareness, a network can gradually identify meaningful physical terms. Below, we present a toy example illustrating how we derive the evolution operator for the Burgers equation. The model is structured as follows: 1) A Fourier Layer; 2) A residual connection to a convolution layer; 3) A ProdLayer. Visualizing the weights of the Fourier layer reveals terms like $k$ and $k^2$, which correspond to the first and second derivatives obtained through Fourier transforms:

$$\frac{d}{dx}g(x) = \mathcal{F}^{-1}[ik(\mathcal{F}g(x))]; \quad \frac{d^2}{dx^2}g(x) = \mathcal{F}^{-1}[-k^2(\mathcal{F}g(x))]. \tag{9}$$

These two operators correspond to the $-1$ and $-2$ orders of the length dimension. Traditional FNO models struggle with shallower layers. Even with a parameter count increased to $1.25$ million, they often suffer from severe overfitting and fail to learn the actual operator, despite achieving reasonable performance. In contrast, improving FNO with DimOL achieves significantly better results with just

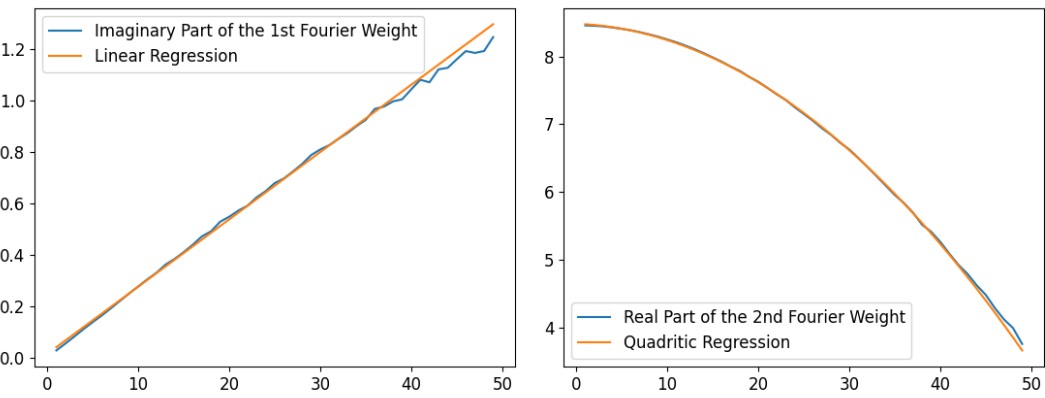

Figure 2: The spectral terms regressed.

Table 7: Sensitivity analysis on various datasets for the hyperparameter choice of $p$ used in Prod-Layer. Results are presented in MSE ($\times 10^{-3}$). For TorusLi, we set the prediction horizon to $T = 10$. For TorusVisForce, we conduct the few-shot experiments with $N = 100$ and $T = 4$. Results marked with an underline indicate the worst performance, while those in **bold** represent the best.

| Dataset | Base | w/o *chan. mix* | $p = 0$ | $p = 1$ | $p = 2$ | $p = 4$ | $p = 8$ | $p = 16$ |
|---|---|---|---|---|---|---|---|---|
| Burgers | T-FNO | 2.225 | 2.342 | 1.130 | 1.147 | 1.163 | **1.059** | 1.071 |
| Darcy Flow | LSM | - | 2.892 | 2.823 | 2.713 | **2.632** | 2.716 | 2.720 |
| TorusLi | T-FNO | 0.1448 | 0.1425 | 0.1396 | 0.1404 | **0.1384** | 0.1398 | 0.1398 |
| TorusVisForce | T-FNO | 2.037 | 2.104 | 1.759 | 1.757 | 1.733 | 1.752 | **1.690** |

a thousand parameters. The minimum MSE on the validation set are $0.0227$ for FNO and $0.0137$ for FNO+DimOL, which means the FNO+DimOL can reduce the error by $39.6\%$.

As is shown in Figure 2, the first Fourier expression we regressed is the imaginary part: $f(k) = 0.02618k + 0.01474$, achieving a Pearson correlation of $0.999$. The second expression, representing the real part, is $-0.001907k^2 + 0.01474k + 8.482$, with a Pearson correlation of $0.9999$.

### 4.8 HYPERPARAMETER SENSITIVITY ANALYSIS

All the experiments presented earlier used a product term number of $p = 2$. What about other $p$ values and models without any channel mixing? In Table 7, we find that all models with ProdLayer at $p = 2$ consistently outperform the $p = 0$ cases (*i.e.*, MLP). The performance does not necessarily improve with the growth of $p$ value; instead, the key factor is whether we employ ProdLayer for channel mixing. Additionally, we observe from Table 7 that if channel mixing with an MLP layer is beneficial, replacing the linear layers with ProdLayer yields even better performance. However, in some cases, such as in the Burgers experiments, adding channel mixing to the original model does not guarantee improvement and may lead to the overfitting problem.

## 5 CONCLUSIONS AND LIMITATIONS

In this paper, we introduced DimOL as an adaptation for major models tailored to solving high-dimensional PDEs. These include FNO-based models and Transformer-based models such as LSM and GNOT. We discovered that dimensional awareness not only makes sense for human physicists but also machine learning models, especially when the task involves actual physical quantities as input. Our research indicates that this conclusion applies across a wide range of models and datasets. By enhancing the model's symbolic meaning, DimOL models are better suited for reverse engineering to uncover terms with potential physical significance. While we currently use ProdLayer for DimOL, we believe that DimOL is more than this. It should be regarded as a new "dimension" to consider when designing AI-for-Science models.

A potential limitation of DimOL could be its effectiveness in tasks where the input data lacks clear dimensional information or where the relationship between dimensions is highly complex or nonlinear. In such cases, DimOL may not offer significant improvements.

### ETHICS STATEMENT

In this work, we adhere to the highest ethical standards across all stages of research. No human subjects were involved, and no personal data was used, ensuring compliance with privacy and security protocols. All datasets utilized are publicly available, mitigating concerns related to sensitive information exposure. We acknowledge the potential use of physics simulation models for harmful insights if misapplied; therefore, we encourage careful consideration of the context and application domain when deploying these models.

### REPRODUCIBILITY STATEMENT

We prioritize the reproducibility of our work. All results can be reproduced by following the experimental details presented in the paper. We provide the source code in the supplementary materials.

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
