# OpenReview forum: "DimOL: Dimensional Awareness as a New 'Dimension' in Operator Learning"
_ICLR.cc/2025/Conference — ICLR 2025 Conference Withdrawn Submission_

### Official Review · Reviewer_hv9q · 2024-10-18

**Soundness:** 2
**Presentation:** 1
**Contribution:** 1
**Rating:** 3
**Confidence:** 4

**Summary:**

The authors introduced DimOL (Dimension-aware Operator Learning) and the ProdLayer to enhance FNO-based and Transformer-based solvers, achieving up to 48% performance gain in 1d Burgers' equation. The new layer increases the ability of the models to handle product terms that appear in many PDEs.

**Strengths:**

(1) The authors introduced a new type of layer called ProdLayer that encodes product terms between different channels/features.

(2) The authors tested the ProdLayer in FNO-based and Transformer-based architectures on several datasets (1D and 2D).

**Weaknesses:**

(1) The paper does not include details about the training process, such as the number of epochs, batch size, learning rate, or the type of optimizer used.

(2) The precise architectures, such as the number of Fourier layers, network width, and number of modes in the FNO case, are not specified. Additionally, it seems no parameter search was performed for any of the experiments.

(3) The models tested have a relatively low parameter count. For example, in the 1D experiment, some models have fewer than 100k parameters.

(4) In the 1D experiment, there is a significant improvement when comparing T-FNO with the ProdLayer to its simpler version. However, this result could be attributed to various factors, such as the possibility that T-FNO training did not converge. Additionally, the improvement is measured against the same model, and even though there is a notable gain of 48.4%, the performance still falls short of unmodified FNO. Therefore, no firm conclusions should be drawn from this gain in a siple 1D experiment. In the 2D case, the improvements are less substantial, and if error bars for the MSE were provided, the gain might fall within the margin of error. In summary, the performance improvement does not seem substantial enough to support strong conclusions.

(5) When the ProdLayer is added to the architecture, the model appears to become more flexible and its training converges more quickly. However, it is unclear whether the models without the ProdLayer were sufficiently trained to make a definitive conclusion about its advantages.

(6) Some parts of the paper are written in an overly informal tone (see, for example, lines 399-402). In line 402, the authors use the phrase, "The datasets are pretty weird by...," which, in my opinion, is not a good example of scientific writing.

(7) No convolution-based baseline models were tested. Some of the state-of-the-art models can be found in [1][2][3].

----

[1] Ronneberger, Olaf, Philipp Fischer, and Thomas Brox. "U-net: Convolutional networks for biomedical image segmentation." Medical image computing and computer-assisted intervention–MICCAI 2015: 18th international conference, Munich, Germany, October 5-9, 2015, proceedings, part III 18. Springer International Publishing, 2015.

[2] Raonic, B., Molinaro, R., De Ryck, T., Rohner, T., Bartolucci, F., Alaifari, R., ... & de Bézenac, E. (2024). Convolutional neural operators for robust and accurate learning of PDEs. Advances in Neural Information Processing Systems, 36.

[3] Gupta, Jayesh K., and Johannes Brandstetter. "Towards multi-spatiotemporal-scale generalized pde modeling." arXiv preprint arXiv:2209.15616 (2022).

**Questions:**

- Are the models without the ProdLayer trained for a sufficient duration to ensure convergence?

- What might explain the significant
performance difference in the Burgers dataset for the T-FNO model? Was the base model trained effectively?

---

### Official Review · Reviewer_iTv3 · 2024-11-03

**Soundness:** 1
**Presentation:** 2
**Contribution:** 1
**Rating:** 3
**Confidence:** 5

**Summary:**

This paper proposes to modify the learning-based PDE solver and claims their modification better fits the perspectives from dimensional analysis. The experiments show that they achieve a marginal gain.

**Strengths:**

No.

**Weaknesses:**

-	The title is misleading. I didn’t see any connection with dimensional analysis, or ‘dimension’.
-	The intuition described in section 3.1 is farfetched. There is certainly sum of products in many physical cases but it does not provide clear motivation on how to derive a new method or ansatz.
In fact, the method proposed by the author is merely a negligible change on the model architecture, having little gain on model expressive power or training.

-	The ansatz (Eq. 5 & 6) is quite arbitrary. There are many different ways to divide features into sub-vectors. It is still unclear what the benefit of this step is.

-	The improvement is negligible. It worth notice that the percentage numbers of ‘gain’ in table 1 and 2 are improvements on MSE instead of MSE itself. For most cases, the gain on MSE is less than 0.5%. It is doubtful whether the gain is larger than the effect of random seeds or other randomness in the experiments and tuning of optimizer hyperparameters and training time, and whether the results shown come from cherry-picking instead of a rigorous investigation.

It is hard to understand why the authors attempted to submit their manuscript to one of the top ML conferences. The quality of the paper does not even meet the bar for a workshop. In recent years, the volume of paper submissions has surged, bringing with it an increasing number of meaningless papers. This is harmful to the whole community.

To conclude, the authors are wasting both their own time and the reviewers’ time.

**Questions:**

.

---

### Official Review · Reviewer_R1Sz · 2024-11-03

**Soundness:** 1
**Presentation:** 2
**Contribution:** 2
**Rating:** 3
**Confidence:** 4

**Summary:**

This paper introduces DimOL (Dimension-aware Operator Learning), a novel approach to enhance neural operator methods for solving partial differential equations (PDEs). The key innovation is incorporating dimensional awareness into neural networks through a proposed ProdLayer that explicitly encodes product terms between channels. The authors demonstrate that this approach can be integrated into various neural operator architectures, including FNO-based and Transformer-based models. The method is evaluated on several standard PDE datasets, showing improvements in prediction accuracy of up to 48% compared to baseline models. The authors also explore the potential of DimOL for symbolic regression and physical term discovery through analysis of Fourier weights.

**Strengths:**

The proposed ProdLayer is a simple yet effective architectural modification that can be easily integrated into existing models with minimal computational overhead. The implementation is clearly described and practically applicable.

The empirical evaluation covers multiple important PDE problems, including both regular mesh settings (Burgers, Darcy Flow, Navier-Stokes) and irregular mesh settings (Inductor2D). The few-shot learning experiments provide interesting insights into the model's generalization capabilities.

The work makes a compelling case for the potential of dimensional awareness in discovering physically meaningful terms through analysis of Fourier components, as demonstrated in the Burgers equation example.

**Weaknesses:**

The theoretical foundation for dimensional awareness appears somewhat limited. While the intuition behind product terms is explained through examples like the heat equation, the paper lacks rigorous analysis of how this generalizes to more complex PDEs, particularly in higher dimensions with spatially varying coefficients.

The empirical improvements, while consistent, are not always substantial. For instance, the improvement on the TorusVisForce dataset over T-FNO (5.5% for T=10) could potentially be within the range of random initialization variance. The paper lacks statistical significance analysis and stability studies across different initializations and hyperparameter settings.

The benchmark problems, while standard, may not fully demonstrate the method's effectiveness for the complex numerical simulations where neural operators are most needed. The paper would benefit from experiments on PDEs with wider parameter ranges (e.g., Reynolds numbers) and more challenging physical systems.

**Questions:**

Could the authors provide statistical significance analysis for the reported improvements, including variance across different random initializations and p-value tests?

How does the method perform on PDEs with wide parameter ranges, particularly for Navier-Stokes equations with varying Reynolds numbers? This would help assess the generalization capabilities of the dimensional awareness approach.

The paper mentions that DimOL can help discover physically meaningful terms. Could this capability be demonstrated on more complex systems beyond the 1D Burgers equation?

What is the theoretical justification for the effectiveness of ProdLayer in cases where the PDE solution cannot be written in a separable form like equation (4)?

Could the authors provide more extensive ablation studies on the architecture choices and hyperparameter sensitivity, particularly regarding the stability of the method?

---

### Official Review · Reviewer_eAqG · 2024-11-04

**Soundness:** 1
**Presentation:** 2
**Contribution:** 1
**Rating:** 3
**Confidence:** 4

**Summary:**

This paper discusses considering dimensional analysis in the neural operator methods, to increase their robustness, accuracy, and interpretability. For this purpose the authors propose ProdLayer, a dimensionally aware neural network layer that can be included in operator learning methods pipelines. The authors provide an intuitive explanation of the proposed methodology, and several experiments comparing operator learning methods with and without the ProdLayer and compare their accuracy and robustness.

**Strengths:**

The paper presents an interesting idea of including dimensional analysis to operator learning, and the benefits of such an approach. Moreover, it provides experiments on standard benchmarks.

**Weaknesses:**

- The most important problem with the paper in my opinion is that the justification of the ProdLayer from the dimensional analysis perspective is really weak and hand-wavy. For example, in Eq. (6) it is assumed that x_2 has the same dimensions as x_a \otimes x_b. It is not explained why this is the case, and the same and is in contradiction with Eq. (5) where x_a, x_b, x_2 have the same dimensions.

- Lines 75-76, 212-213 refer to dimensionality in a different manner, meaning the length of the vector, and not the fundamental dimensions (length, time, etc). Please be consistent, else the paper becomes confusing.

- I believe that it would be beneficial for the authors to work on the grammar, because there are sentences and paragraphs that do not read well. For example, lines 29-38, 97-101, 115-117, 125-127, 135-138, to provide a few examples.

- Eq. 2 refers to the momentum balance of the Incompressible Navier-Stokes for Newtonian Fluids. Please be specific about which problem you refer, or else provide the general definition.

- The results are presented in the MSE loss, which is not a relative error metric and therefore not informative. Please use an error metric such as the relative L1 or L2 errors.

- The statement in 103-105 about the two branches of Operator Learning is wrong. Please remove it.



- The gain of using ProdLayer is really small and I do not believe that it is statistically significant.

**Questions:**

- Could you please further explain the choice behind the ProdLayer definition as a product and a sum.

- Could you explain what you mean by polynomial structured terms in line 189? Do you mean terms such as L/T^2 that describe dimensions?

- Is the Latent Spectral Method something you propose or something that others have done? Could you provide more information or a citation on that ?

- Could you please further explain Eq. (7)?  You state in line 454 " This property .... traditional dimensional analysis". However, it is not clear from Eq. (7) is k has dimensions or not.

- Can you provide a citation or explain what the "invariance of similar transformations" is?

---

### Note · Authors · 2024-11-13

I have read and agree with the venue's withdrawal policy on behalf of myself and my co-authors.